Journal of Data-centric Machine Learning Research (2024)        Submitted 5/24; Revised 8/24; Published 9/24

# ComPile: A Large IR Dataset from Production Sources

**Aiden Grossman**[+,*,1]                                                  AMGROSSMAN@UCDAVIS.EDU
**Ludger Paehler**[*,2]                                                        TWO@CS.BERKELEY.EDU
**Konstantinos Parasyris**[3]                                               PARASYRIS1@LLNL.GOV
**Tal Ben-Nun**[3]                                                               TALBN@LLNL.GOV
**Jacob Hegna**[4]                                                        JACOBHEGNA@GMAIL.COM
**William S. Moses**[5]                                                     WSMOSES@ILLINOIS.EDU
**Jose M. Monsalve Diaz**[6]                                          JMONSALVEDIAZ@ANL.GOV
**Mircea Trofin**[7]                                                         MTROFIN@GOOGLE.COM
**Johannes Doerfert**[*,3]                                                   JDOERFERT@LLNL.GOV

[1] *University of California, Davis, USA*

[2] *School of Computation, Information and Technology, Technical University of Munich, GER*

[3] *Center for Applied Scientific Computing, Lawrence Livermore National Laboratory, USA*

[4] *University of Minnesota, Twin Cities, USA*

[5] *Department of Computer Science, University of Illinois Urbana-Champaign, USA*

[6] *Division of Mathematics and Computer Science, Argonne National Laboratory, USA*

[7] *Google Inc., USA*

**Reviewed on OpenReview:** *https://openreview.net/forum?id=abc3*

**Editor:** Sebastian Schelter

## Abstract

Code is increasingly becoming a core data modality of modern machine learning research impacting not only the way we write code with conversational agents like OpenAI's ChatGPT, Google's Bard, or Anthropic's Claude, the way we translate code from one language into another, but also the compiler infrastructure underlying the language. While modeling approaches may vary and representations differ, the targeted tasks often remain the same within the individual classes of models. Yet, relying solely on the ability of modern models to extract information from unstructured code does not take advantage of 70 years of programming language and compiler development by not utilizing the structure inherent to programs in the data collection. This detracts from the performance of models working over a tokenized representation of input code and precludes the use of these models in the compiler itself. To work towards the first intermediate representation (IR) based models, we fully utilize the LLVM compiler infrastructure, shared by a number of languages, to generate a 1.4T Llama 2 token dataset of LLVM IR. We generated this dataset from programming languages built on the shared LLVM infrastructure, including Rust, Swift, Julia, and C/C++, by hooking into LLVM code generation either through the language's package manager or the compiler directly to extract the dataset of intermediate representations from

---

+. Work performed while at LLNL

*. Corresponding authors

. Permissibly licensed subset of the dataset available under huggingface.co/datasets/llvm-ml/ComPile

production grade programs. Statistical analysis proves the utility of our dataset not only for large language model training, but also for the introspection into the code generation process itself as well as for training of machine-learned compiler components.

**Keywords:** Code, Multilingual, Compiler, LLVM, Intermediate Representation

## 1 Introduction

With the encapsulation of attention (Chorowski et al., 2015) in the modern transformer architecture (Vaswani et al., 2017), the transformer has dominated many natural language processing tasks, starting with the widely used BERT architecture (Devlin et al., 2018). Adjacent fields, such as vision (Alayrac et al., 2022), and cross-modal models, such as natural language to vision models, all have been transformed by the modern singular architecture approach. Originally, due to the immense computational cost of training (Hoffmann et al., 2022), models with full weights for further training were only sparsely available. Fine-tuning for downstream tasks (Brown et al., 2020), or task suites (Srivastava et al., 2023), allows modern large language models to solve a wider array of modeling tasks. In the past few years, there has been a Cambrian explosion in the availability of pre-trained capable models for fine-tuning. There are currently a large number of open-source release of pre-trained model series such as OPT (Zhang et al., 2022), Llama (Touvron et al., 2023a), Llama 2 (Touvron et al., 2023b), Pythia (Biderman et al., 2023), MPT Portes et al. (2023), and the recently released StarCoder 2 (Lozhkov et al., 2024). The wider availability of model weights, architectures, and training checkpoints has enabled the application and tuning of these increasingly capable large language models for domains such as code.

Beginning with the first BERT models for code (Feng et al., 2020) and their extension to graphs (Guo et al., 2020), code has remained a highly active data modality and has seen a constant flurry of new ideas, interfaces, representations, and downstream tasks. Most recently, the rise of instruction-tuned (Ouyang et al., 2022), and reinforcement learning-trained large language models (Christiano et al., 2017) have enabled a completely new interface to these models. For example, by conversing with a large language model for code (OpenAI, 2023a; DeepMind, 2023), the user *prompts* the model with their query, and the model then writes the prompted-for code. This has, in turn, spawned an entire class of new prompting approaches specifically designed for code, such as grammar-based sampling and sequential Monte-Carlo steering (Lew et al., 2023). Approaches to the construction of large language models for code vary. Some large language models use a base model, such as, e.g., Code Llama (Rozière et al., 2023), that is fine-tuned through a general training corpus that only contains code (OpenAI, 2023b; Anthropic, 2023). On the other hand, other models are only trained on code from the outset (GitHub, 2023; Roziere et al., 2021; Szafraniec et al., 2023). We focus on the category of models for which one utilizes a pre-trained base building block trained on a larger training corpus consisting of not only code, to then fine-tune on a code-only *code training* corpus. Recently, several models have appeared, such as Meta's Code Llama (Rozière et al., 2023), Alphabet's Codex (AI, 2023), WizardCoder (Luo et al., 2023), and the large cross-institutional collaborations StarCoder (Li et al., 2023; Lozhkov et al., 2024), and SantaCoder (Allal et al., 2023). All these share the goal of assisting people in writing code but miss the opportunities afforded by combining the properties of these powerful modern models (i.e., scale, capabilities, and adaptiveness) with the established

shared compilation infrastructure that has made programming languages faster, more robust, and easier to use.

In several pieces of previous work (Haj-Ali et al., 2020; Kulkarni et al., 2013), this transformative potential was harnessed, machine-learned heuristic replacements developed, and in some cases (Trofin et al., 2021) the *heuristics were upstreamed to the main LLVM codebase*, improving all code run through LLVM when the ML heuristics are enabled. Orthogonal to the replacement of heuristics with machine learning, a large number of people have explored the ordering of compiler passes (Cummins et al., 2022; Huang et al., 2020). While the learning of pass orderings was initially held back by the lack of easy-to-access, high-performance reinforcement learning environments to validate new reinforcement learning strategies, this has by now been addressed with the introduction of CompilerGym (Cummins et al., 2022). In contrast, the learning of entirely new heuristics, optimization passes, and other compiler components with large language models (Yang et al., 2023; Cummins et al., 2023) to realize the transformative potential of the model class is held back partially by the lack of large datasets of high-quality code to train such models properly. Models are only trained on smaller datasets, such as Anghabench (Da Silva et al., 2021), Exebench (Armengol-Estapé et al., 2022), and HPCORPUS (Kadosh et al., 2023a), or sometimes rely on synthetic benchmarks. Synthetic benchmarks can be aided by ML techniques (Cummins et al., 2017; Tsimpourlas et al., 2023) and even closely match some properties of the corpora they are trained on, but these techniques themselves still suffer from a small training set and can only approximate the properties of production code. Small datasets lead to smaller, worse-performing models and hence do not allow such compiler-focussed models to fully access the fine-tuning paradigm utilized by modern large language models with their base models of multiple billion parameters.

## 1.1 Contributions

Focusing on the paradigm of taking a pre-trained basic building block, we pose the question *"What does a modern, large code training dataset for compilers actually look like?"* and construct a high-quality dataset of a similar scale to existing LLM datasets solely at the level of compiler intermediate representation. Within this context, we associate quality with the *usage* of code, with code being used more often being of higher quality for our purposes. Correctly being able to reason about very widespread code in production systems is incredibly important for compiler work. In the short term, we believe our dataset will enable the training of larger language models for compilers useful for a broader array of downstream tasks after fine-tuning, and in the long-term enable use-cases such as direct performance prediction to obtain a reliable runtime estimate without ever needing to run a single line of code. To these goals, our paper makes the following contributions:

- The **introduction of a 2.8TB dataset** of textual intermediate representation of the shared LLVM compiler infrastructure encompassing **production-grade programs from Rust, Julia, Swift, and C/C++**. A broad overview of the size distribution is shown in Figure 1.

- Preliminary **large scale statistical analyses of LLVM-IR modules across multiple languages**, demonstrating the utility of our dataset and tooling.

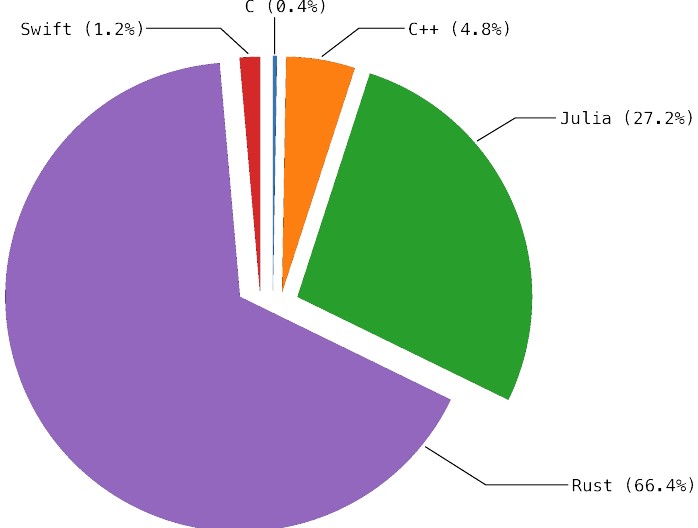

Figure 1: Size distribution of LLVM intermediate representation (IR) bitcode within ComPile before de-duplication within and among languages.

- Demonstration of the utility of ComPile for training large machine learning models through **quantification of size and approximate token counts**.

- **Open-sourcing of workflow and compiler tooling** to construct massive-scale code datasets, which are easy to install and ready for scalable deployment in HPC and cloud environments. The statistics of the entire dataset constructable with the tooling are available in the appendix A.

| Language | | Llama 2 Tokens *(billions)* | BPE Tokens | | | |
|---|---|---|---|---|---|---|
| | | | 10k *(billions)* | 50k *(billions)* | 50k *(billions)* | 100k *(billions)* |
| | C | 5 | 1 | 1 | 0.5 | 0.4 |
| | C++ | 47 | 11 | 6 | 5 | 4 |
| | Julia | 548 | 42 | 23 | 18 | 12 |
| | Rust | 736 | 137 | 90 | 79 | 69 |
| | Swift | 20 | 3 | 2 | 1 | 1 |
| **Total** | | **1355** | **195** | **122** | **104** | **88** |

Table 1: Token count of the encoded ComPile under varying vocabulary sizes, and considering the tokenization of the data with Byte-Pair encodings *(BPE)*, and tokenization with the Llama 2 tokenizer.

| Language | Source Code | Unoptimized IR | Optimized IR | X86 Assembly |
|----------|-------------|----------------|--------------|--------------|
| C | ```int sum(int a, int b)
{
    return a+b;
}``` | ```define i32 @sum
(i32 %0, i32 %1) {
  %3 = alloca i32
  %4 = alloca i32
  store i32 %0, ptr %3
  store i32 %1, ptr %4
  %5 = load i32, ptr %3
  %6 = load i32, ptr %4
  %7 = add i32 %5, %6
  ret i32 %7
}``` | ```define i32 @sum
(i32 %0, i32 %1) {
  %3 = add nsw i32 %1, %0
  ret i32 %3
}``` | ```sum:
  push    rbp
  mov     rbp, rsp
  mov     eax, esi
  add     eax, edi
  pop     rbp
  ret``` |
| Rust | ```pub fn sum(
    a: i32, b: i32
) -> i32 {
    a + b
}``` | ```define i32 @a::sum
(i32 %a, i32 %b) {
start:
  %_0 = add i32 %a, %b
  ret i32 %_0
}``` | ```define i32 @a::sum
(i32 %a, i32 %b) {
start:
  %_0 = add i32 %a, %b
  ret i32 %_0
}``` | ```example::sum:
  mov     eax, edi
  add     eax, esi
  ret``` |

Table 2: The transformations source code goes through into assembly through the compiler's LLVM intermediate representation. We collect the intermediate representation at the unoptimized stage.

## 2  Background

Building upon package ecosystems as sources of intermediate representation is ideal due to the large amount of packaged high-quality code and the abstraction over the build systems of individual projects. This abstraction is due to a common build wrapper that invokes the individual build systems with the relevant configuration options. Package managers are designed to install a set of packages that a user desires, abiding by some constraints from some repositories. Each package manager often has its own repositories that are built from source. The recipes used to build the included applications often specify exact build steps to build a piece of software, including an exact and consistent specification of dependencies needed to build said software. These package recipes can also often be modified to perform some additional steps or to modify the build process itself. Some build systems, such as cargo, are combined with package managers, allowing them to build a piece of software and all of its dependencies that the build system supports installing itself. Modifying these build processes allows us to take advantage of the dependency management and other aspects of build recipes already present for a significant number of packages. However, many build systems do not explicitly support custom modification of build recipes or build-time configuration options, including compile flags. In this work, we choose to specifically focus on utilizing package managers that explicitly allow setting compiler flags, such as the from-source package manager Spack (Gamblin et al., 2015) that is focused on high-performance computing (HPC).

In addition to utilizing package managers, we also take advantage of several aspects of the LLVM compilation infrastructure (Lattner and Adve, 2004), particularly the Clang C/C++ frontend and LLVM-IR, the intermediate representation LLVM uses. The full process of compilation, such as the one performed by Clang with LLVM during the compilation of C/C++, is composed of three main stages: the frontend, the middle-end, and the backend.

The entire compilation process is exemplified in Table 2. A compiler frontend has the job of taking a piece of source code, typically a single source file, sometimes called a translation unit, and generating a *module* of intermediate representation that can then be processed by a compiler middle-end, such as LLVM. A module typically contains multiple functions, referenced globals, and relevant metadata. Compiler intermediate representations, or IRs, are designed to sit between the source programming language and the compiler's output, assembly. They are typically designed to be source-language and target-agnostic. This allows code written to modify and process the IR to be reused across many languages and target platforms. IRs typically also have additional properties that make them particularly amenable for performing optimizations. LLVM-IR specifically enforces single static assignment (SSA), where all variables are assigned exactly once and referenced multiple times. This makes certain analyses much easier, such as dataflow analysis. LLVM uses its intermediate representation, LLVM-IR, to perform optimizations and other operations related to lowering source code to machine code in a manner that abstracts away most details of the target machine. Within LLVM, the compiler middle-end operates over the IR produced by the frontend through a series of grouped operations called passes. A *pass* is designed to perform a specific task, such as removing dead code, simplifying the control flow graph, or combining instructions that can be simplified. A *pass pipeline* is typically language and optimization-level specific. It comprises a set of passes in a specific order run over the IR to optimize it for the desired properties. After optimization, the compiler backend takes over, performing the necessary tasks to transform the (mostly) target-agnostic IR into target-specific machine code that can be executed on the target machine. The backend typically performs tasks such as instruction selection, instruction scheduling, and register allocation. In addition, compiler backends also often perform some small target-specific optimizations, such as peephole optimizations, to further improve the characteristics of generated code. We compose our dataset, *ComPile*, of LLVM-IR, as it gives a common framework across programming languages and target platforms while also allowing us to perform a detailed analysis of the compiler middle-end. These properties and more make LLVM-IR a great modality for a compiler-centric dataset useful for compiler tasks such as program analysis, optimizations, and code generation.

## 3 Dataset Construction

The building of entire language ecosystems introduces its very own kind of problems, such as "How can we build all packages while only manipulating a single builder file?", "How can we distribute the build process of all these packages across many nodes?" and in the case of just in time (JIT) compiled languages "How well defined is the compilation of entire packages?". In this section, we describe in detail our workflow and all modifications to the build systems of individual languages. A summary of our workflow can be seen in Figure 2. To construct the IR database, we use a set of curated sources focusing on code used in production systems. Individual sources are defined in `.json` files. While most projects are hosted in repositories on GitHub, we also added sources consisting of archived compressed source codes such as tarball files. The builders then ingest the information from the project on its build system, either through the manifest information, which contains the information on the building mechanism and commands, or through an ecosystem specific manifest processed by a script into a complete package manifest. Next in the workflow is

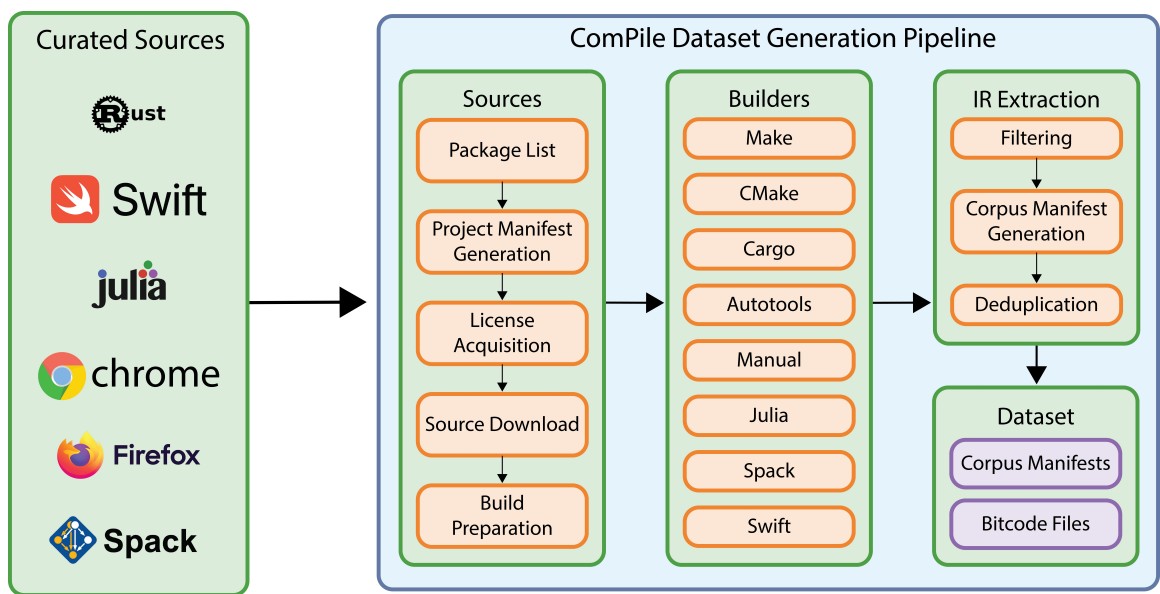

Figure 2: Individual components of the dataset collection tooling. *(Curated Sources)* The set of sources comprised of package indices, and selected packages, ingested by the ComPile Dataset Generation Pipeline. *(Sources)* acquire the source based upon the provided package list, before the *(Builders)* built the package, and it is then filtered, deduplicated, and its build process documented in the *(IR Extraction)* to arrive at the dataset.

the LLVM-IR extraction. Extracting IR depends on the way the IR is presented in the curated source, as we will be described below. A deduplication stage removes exact or near duplicate IR to eliminate common IR, or IR that does not improve the quality of our corpus, which has established precedent in literature (Allamanis, 2019). A manifest that contains a list of LLVM bitcode modules extracted from the project is then created. Leaning into the shared LLVM compiler infrastructure, we are able to take advantage of existing LLVM tools and LLVM passes to obtain information about the LLVM-IR modules. After building, IR extraction, and deduplication, the dataset is then ready for downstream usage in analysis or training capacities. The source-, and language-specific builders, LLVM IR extraction, deduplication, and filtering are packaged into a pip-installable Python package, which is readily available on PyPi [1], and Github [2]. It is maintained adjacent to the wider LLVM organization, and is aligned with other LLVM efforts around the use of machine learning in compilers.

---

1. pypi.org/project/llvm-ir-dataset-utils/
2. github.com/llvm-ml/llvm-ir-dataset-utils

### 3.1 Ecosystem-Specific Builders

We support extracting LLVM-IR from several large package ecosystems through many different builders that each handle a specific ecosystem or build system. The major builders that have been implemented are described below.

#### 3.1.1 Rust

For extracting IR from Rust packages, we first extract a list of crates from the Rust Package Repository (rus, 2023). The Rust Package repository has over 100,000 Rust packages that are all buildable using a consistent build system, `cargo`, which makes the build process very feasible to automate. Additionally, the process of listing packages in the repository serves as a filter, preventing the ingestion of unused or experimental Rust code, which could otherwise feasibly impact the IR distributions within the final corpus. We prioritize building crates from the indicated git repository in the package repository as we found that git repositories often have additional targets not included in the tar archives uploaded to the package repository that yield more IR. We remove crates that point to the same git repository to improve build times and prevent excessive duplication within the dataset. In addition to pulling from a git repository, we also use the package repository provided tar archive as a fallback as we found that many git repositories would fail to clone. For building each crate, we used the native cargo build system. We first extract a list of targets and sub-packages from the package manifest and then built each one with `cargo rustc`, specifying the appropriate sub-package and target and passing the `--emit=llvm-bc` flag to make cargo additionally generate LLVM IR. We built Rust packages without any optimizations to ensure we get unoptimized bitcode. This could feasibly impact the distribution of our dataset as Rust has two high level Rust-specific IRs that are used to optimize Rust code before it is lowered into LLVM-IR that do not perform optimizations without optimization flags. We leave analysis of distribution shifts related to pre-LLVM optimizations to future work.

#### 3.1.2 Julia

To extract code from Julia, we used the official package registry (JuliaLang, 2023a) as a source of over 9000 packages. We then processed them using a custom pipeline to extract bitcode. Due to the nature of Julia's "ahead-of-time just-in-time" (AOT-JIT) design, the "compilation" of a whole package is an ill-defined task. Julia only completely lowers a function to IR at runtime when the function is called. This makes the extraction of bitcode for an entire package difficult as every single function within a package has to be run with all potential function signatures. However, there has recently been a large push to precompile packages using `PackageCompiler.jl` (JuliaLang, 2023b). The precompilation process involves completely compiling a variety of often used functions within a package and caching them for later use so the user experiences less wait time when using the package later. This is performed automatically for many packages during the installation process. In addition to taking advantage of IR generated during the precompilation process, we also run unit tests, if available, to force the lowering of additional functions. To grab bitcode per package, we implemented a custom hook in the compilation process[3]. We subsequently post-

---

3. github.com/JuliaLang/julia/pull/50946

process the produced files to only grab IR which contains actual code rather than serialized Julia data structures. All the IR files for a package are gathered with the dependencies to capture all uses of a function. This results in duplicate code, which is removed through a deduplication pipeline. This process inevitably leaves some gaps in the collected IR. For example, we are not capturing the function specializations exactly used in production code, instead capturing the function specializations deemed important by the package authors in the package compilation step and from the available unit tests. However, we believe these nuances of the collection process do not impact the results presented in this paper or the utility of the collected Julia code significantly.

### 3.1.3 SWIFT

To prepare a list of Swift packages, we used the Swift Package Index (Community, 2023), processing all the GitHub repositories present in the packages list. We then cloned and built every repository we found with Swift. Swift automatically resolved the dependencies that it was able to and, by passing the flags `--emit-swift-module-separately -Xswiftc --embed-bitcode`, we are able to embed bitcode within the object files produced during the compilation process for extraction later. To embed the bitcode correctly, we have to use the `--emit-swift-module-separately` flag to deactivate the default behaviour of `swiftc` to emit partial modules, and only merging them later, which is incompatible with bitcode embedding. While we were able to get some packages to build, our tooling is designed for a Linux environment, and not the Swift-preferred MacOS environment. Although Linux has Swift platform support, it does not have support for several of the closed-source dependencies that many Swift packages require such as SwiftUI.

### 3.1.4 SPACK

To include HPC packages, we utilized the HPC package manager Spack (Gamblin et al., 2015). Spack contains a large set of packaged applications, many of them C/C++, and it lets the user specify the compiler toolchain and any compiler flags to use. In addition, the packaging process in Spack serves as a quality filter for the dataset, as Spack selects for only those HPC packages whose developers or users have opted to take steps to contribute their software to Spack. Getting a package into Spack requires review on GitHub, and this tends to select for popular HPC packages that people *want* to use. While Spack also contains packages that use a significant amount of Fortran, we only extract bitcode generated from C/C++, because most Fortran packages are not yet compatible with recent LLVM-based Fortran frontend.

To build a corpus of IR from Spack, we start by extracting a list of packages. Spack supports a variety of different package types, including Python packages and custom build systems, many of which will not produce IR. We filter the packages by build system, including only packages that use the common C/C++ build systems CMake, Meson, Autotools, and Makefiles. After we have a list of packages, we then *concretize* each package. *Concretization* is the process of generating the fully satisfied dependency graph (including flags, build options, microarchitecture targets, and optional dependencies) for each package. Spack can optionally unify the dependency graph for packages to ensure that each dependency is built only once, in one configuration, but we choose to concretize each package separately to allow

each package to have its own dependency configuration. This aids in error handling, and it allows us to run the concretization process in parallel. If any package is incompatible with another for a *unified* set of packages, concretization fails. If we allow packages to have their own dependency versions, we can split up the process and handle individual failures more gracefully. However, this methodology leaves us with many duplicate packages in addition to dependencies that won't produce any bitcode. We handle this by building all packages in the same manner and passing all extracted bitcode through a deduplication pipeline.

After concretization, we build all packages with a custom build distribution system, starting with leaf dependencies and continuing on as more and more packages have all of their dependencies built. We build each package with `clang` while passing the compiler flags `-Xclang -fembed-bitcode=all`, which causes LLVM IR to be embedded as bitcode within the generated object files. To extract IR from built packages, we direct Spack to keep the build directory (which contains .o files, libraries, and other build artifacts) by passing the `--keep-stage` flag. To allow for multi-node parallelism, we take advantage of Spack's buildcache feature, pushing all built packages to a buildcache so that any node within an allocation can use a built package as a dependency. This allows us to distribute builds across a large cluster and obtain a high degree of parallelism, significantly reducing overall build time for the corpus.

### 3.1.5 Individual Packages

In addition to collecting a significant number of packages available through specific ecosystems, we also wrote additional tooling to allow for the collection of bitcode from individual curated packages. We wrote scripts to build applications that use CMake, Autotools, and any other build system that can be invoked through raw shell commands. These scripts work by invoking the build system using the user-provided arguments along with some additional flags. These additional flags include setting the compiler to `clang` to make bitcode extraction possible and passing `-Xclang -fembed-bitcode=all` as C/C++ flags to ensure that bitcode was inserted into the generated object files. The bitcode is then extracted from the object files after the build completes which is available for further analysis. We collected bitcode from several large applications not included in the existing package ecosystems that we deemed to be high impact including Chromium, Firefox, and the Linux Kernel. These programs each consist of upwards of tens of gigabytes of bitcode and contain production code that is run virtually everywhere.

### 3.2 LLVM-IR extraction

The aim of our IR extraction approach is to extract IR immediately after the frontend, before any LLVM optimization passes have run. This allows us to perform analysis on the IR emitted directly after the frontend, and anywhere else in the optimization pipeline, as we can perform optimization manually, and introspect the optimization pipeline itself. It is important to note that some languages like e.g. Rust and Julia use language-specific higher level intermediate representations for optimizations and other transformations specific to the specific language that we are not able to introspect with our approach. The process for extracting IR directly after the frontend differs significantly depending upon the language with the necessary options and configurations for doing so being reported in the build

| Programming Language | Bitcode *(GB)* | Deduplicated Bitcode *(GB)* | Licensed Bitcode *(GB)* | Licensed Text *(GB)* |
|---|---|---|---|---|
| C | 16 | 8 | 2 | 10 |
| C++ | 109 | 74 | 29 | 103 |
| Julia | 200 | 184 | 164 | 1088 |
| Rust | 656 | 580 | 400 | 1524 |
| Swift | 8 | 7 | 7 | 36 |
| **Total** | **990** | **853** | **602** | **2761** |

Table 3: Amount of bitcode contained in the public version of ComPile before and after deuplication, and the size of the bitcode and associated textual IR for the public version of ComPile.

processes above. After the build process completes for a package we are left with an assortment of bitcode in two different formats depending upon the build system: bitcode embedded in object files or a collection of separate bitcode files. To extract the bitcode into a structured corpus, we take advantage of the `ml-compiler-opt` tooling from MLGO (Trofin et al., 2021) for the extraction of IR object files by analyzing a structured compilation command database, or alternatively by searching for all object files within the build directory. In addition, it also supports creating a structured corpus from raw bitcode files by searching the build directory. The exact strategy used is dependent upon the build system. Julia, and Rust directly emit bitcode. Spack, CMake, Autotools, and manual builds are all currently set up to embed bitcode in object files, but only CMake is able to provide a structured database of compilation commands. Swift embeds bitcode but needs additional flags during IR extraction due to the bitcode section naming within the object file differing from clang's. During IR-extraction we do not strip debug information if it is present as it can easily be stripped later and some models need to have some debug information in their training corpus to be robust against it. Some builders emit debug information more commonly than others, such as Rust where we compile in debug mode by default to disable optimizations, but ultimately whether or not debug information is present is project dependent. Finally, we specifically collect the compressed bitcode rather than textual IR, which can easily be converted into the latter by running `llvm-dis` over the collected corpus.

### 3.3 Deduplication

Training dataset deduplication is important for model performance (Allamanis, 2019; Kandpal et al., 2022). To this end, we deduplicate the entire dataset presented in this paper at the module level by computing a combined hash of all global variables and functions. To perform the hashing, we upstreamed the `StructuralHash` to LLVM through the `StructuralHashPrinterPass` [4] [5] [6]. The structural hashing process only captures semantic

---

4. reviews.llvm.org/D158217
5. reviews.llvm.org/D158250
6. reviews.llvm.org/D158317

details of the IR making it invariant against all changes that do not impact the meaning of the IR other than function call names. In addition, the implementation does not capture all semantic information, currently ignoring details such as attributes and instruction dependencies which ensures that near-duplicates may be matched as well. We chose to deduplicate at the module level as this ensures the majority of the duplicate code is removed from the dataset while leaving all significant context within each module for performing module-level tasks. This deduplication strategy prevents some tasks from being performed, such as project-level tasks, which rely on a complete set of modules or metadata. The degree of deduplication is heavily language dependent. In the case of Julia for example, a duplicationr rate of $\approx 40\%$ is found due to the inclusion of bitcode from package dependencies. Other languages have significantly lower duplication rates.

### 3.4 Dataset Size

To analyze the size of the dataset, we directly gather the size of all bitcode files in the corpus before and after deduplication. A 35% reduction in dataset size after dedupliction is observed. While measuring the size of bitcode files gives some idea of the total size, it does not allow for proper size comparisons to other datasets as LLVM bitcode is highly compressed. To this end, we also compute the size of all textual IR in the dataset by measuring the size of all disassembled bitcode files. We find that the size of textual IR is approximately 4.6 times the size of the equivalent bitcode. Precise size figures for ComPile are available in Table 3.

### 3.5 License Filtering

To filter our closed-source dataset for permissively licensed projects, we filter the entire database of projects compiler into ComPile for the `MIT`, `Apache-2.0`, the `BSD-3-Clause`, and the `BSD-2-Clause` licenses. For this we obtain the license information from package repositories, GitHub, and in part manually using the `go-license-detector` [7], and distribute provenance information, and license text along with the dataset to comply with terms.

## 4 Statistical Analysis

To characterize the dataset, its inherent statistical utility, and its utility for the training of large language models, a number of statistical analyses are performed. The ability to explore, and compare these analyses cross-language is a core novelty of our dataset to compiler engineers, as well as to the construction of machine-learned compiler componentry.

### 4.1 Visualization of Properties

The function properties are computed using the upstream `FunctionPropertiesAnalysis` pass in LLVM, which we modified [8] [9] [10] to give us a similar set of features to YaCoS (Filho et al., 2018). To better understand the characteristics of the collected IR in terms of features of the underlying source code, and of the language itself, several analyses are performed.

---

7. `https://github.com/go-enry/go-license-detector`
8. reviews.llvm.org/D157358
9. reviews.llvm.org/D158018
10. reviews.llvm.org/D158681

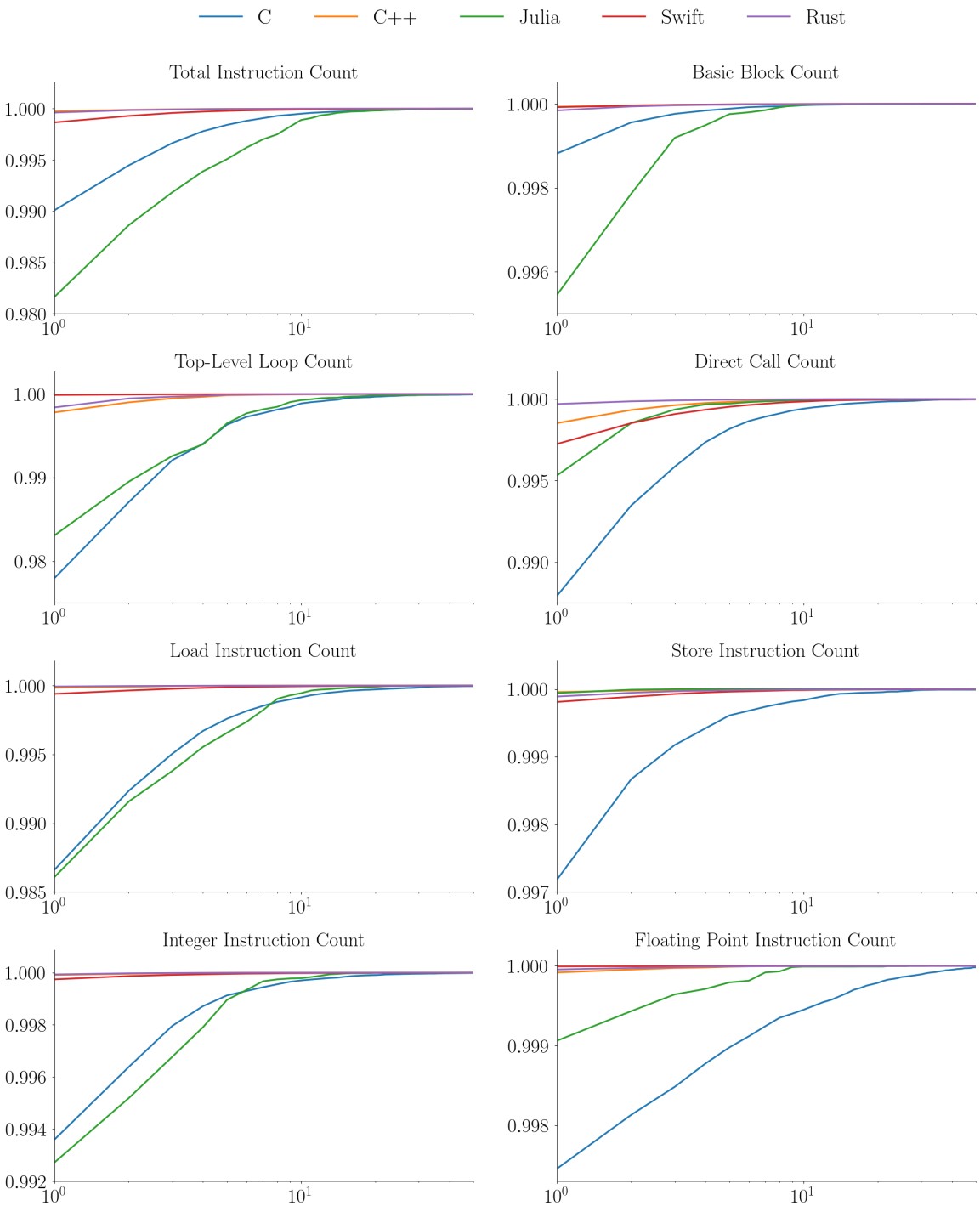

Figure 3: Cumulative distribution functions in which the function properties are first converted into probability density functions, before being plotted cumulatively. All function properties are analyzed across all 5 languages, and show a similarly strong left-skew in their count-statistics.

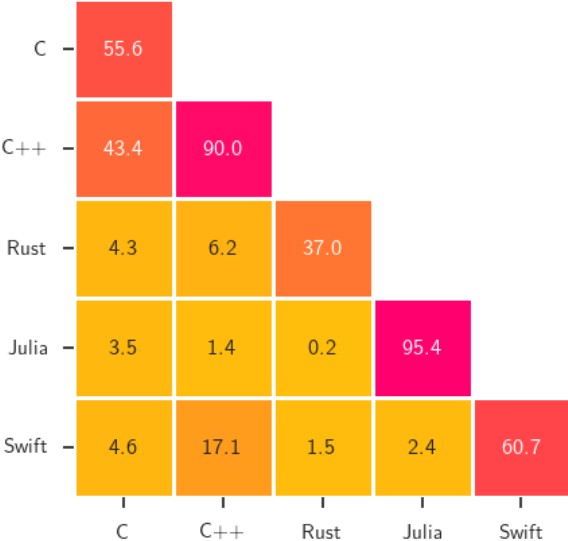

Figure 4: Percentage of duplicate functions present between two languages as determined by the newly upstreamd `StructuralHash` in LLVM with detailed hashing enabled. All values are percentages.

## 4.2 Function Properties

In addition to looking at the combination of function properties, we also looked at several function properties individually, comparing them across languages as shown in Figure 3. We collected properties using LLVM's `FunctionPropertiesAnalysis` pass, sampling 1,000,000 functions from each language contained within the dataset. All of these variables show the same overall shape, a strongly left-skewed distribution, but the exact characteristics are language dependent. In addition, most of these properties are correlated with the length of the function under analysis, but show some distinct patterns depending upon the variable under analysis such as the load instruction count and the floating point instruction count where certain languages have a significantly longer tail than other languages, C and C++ for load instructions, and C for floating point instructions. There are several other patterns such as the significantly longer tail for C++ in regards to direct calls, suggesting small-function idioms. The long-tail of top-level loops in C/C++ also suggests some information about their usage.

## 4.3 Function Duplication

Furthermore, we performed an analysis quantifying function duplication within and between languages and present our results in the heatmap shown in Figure 4. To compute function duplication, we used a similar methodology to the one used for module-level deduplication in the initial deduplication stage. We deduplicated using LLVM's `StructuralHash`, but for this analysis we looked at individual functions rather than the whole module. Within the deduplicated data, some interesting patterns emerge. There is a much higher degree of

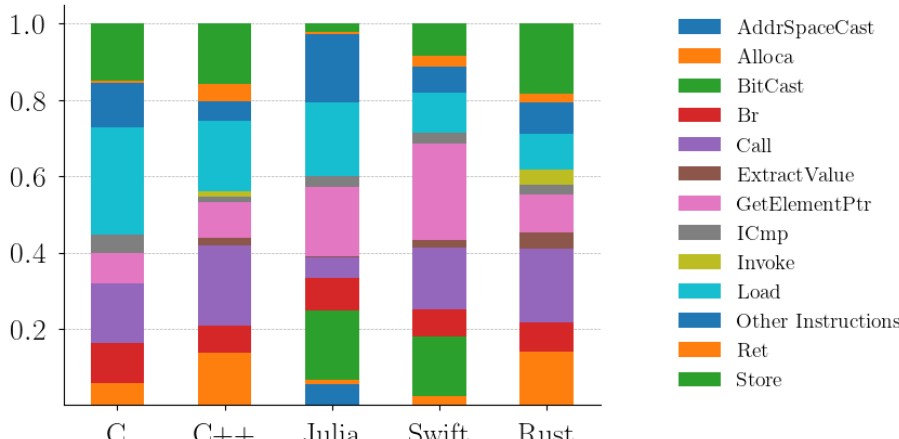

Figure 5: LLVM IR opcode distribution of the top ten operations across all languages included in ComPile as computed by LLVM's `InstCount` pass.

duplication within individual languages than there is between languages. We hypothesize this is caused primarily by the following two factors: language idioms and function mangling. There are often a significant number of idioms within a language such as getters and setters in C++ that will often end up producing similar IR, causing a high degree of duplication within a language. In addition to this, different languages use different mangling strategies, which significantly decreases the duplication rate between languages for functions that involve function calls as `StructuralHash` takes function names into account when evaluating call instructions, on top of the names of the called functions potentially being different. However, we do see more duplication between languages that share similar niches and compilation strategies. There is a significant amount of overlap between C and C++ as they occupy similar software niches and, when compiled with clang, share a compiler frontend and middle end. We also see some overlap between C++ and Swift, suggesting some similarity, potentially in language idioms. Next, we observe that there is virtually no overlap between Julia and other languages, again supporting the hypothesis that Julia emits code significantly different from other languages. Finally, we observe duplication between Rust and C++ in addition to a smaller amount of duplication between Rust and C, but both are quite small.

### 4.4 Opcode Distribution

Next, we perform an analysis of the instruction distribution across languages. We use the LLVM `InstCount` pass to count instructions at the module level and then aggregate the total number of instructions per language. This pass ignores extraneous instructions like debug instructions, but does count some LLVM annotations presented as intrinsics, such as lifetime annotations, as call instructions. There are several interesting differences between the frontends for various languages that we observed. For example, we observed that Julia emits significantly less store instructions than other languages, but takes significant advantage of instructions not within the ten most frequent instructions compared to other

languages. Other differences, such as the significant number of return instructions in C++ suggest a large amount of small functions, which we also see to a lesser degree in Swift and Rust, which share some OOP idioms. However, this could also be due to multi-exit functions as a choice of the program or as a pattern within the source code. In addition, we see that certain languages are the only ones to use specific instructions. For example, Julia and Swift make extensive use of the `BitCast` instruction while C, C++, and Rust do not use it. Finally, we observe an increase in call instructions after optimization, which is most likely a result of inlining. It is important to note that an increase in call instructions only describes the static count and the number of runtime call instructions would likely decrease.

### 4.5 Token Count

Finally, we performed experiments at different vocabulary sizes to gather approximate token counts to determine the utility of our dataset for the code training of pre-trained large language models with the results shown in Table 1. For these experiments we used the Llama 2 tokenizer (Touvron et al., 2023b) to be able to compare ComPile's size to contemporary datasets. To further test the size of the dataset, we generated a vocabulary from a subset of the dataset. We chose to use BPE tokenization (Sennrich et al., 2016) as it is one of the most commonly used techniques for tokenization for LLMs and easily adaptable to the textual component of our dataset. We gathered approximately 400 bitcode modules from each language and disassembled them into IR, training a BPE tokenizer over this data using fastBPE [11], generating several different vocabulary sizes for the various experiments. Finally, we used fastBPE to tokenize all the modules in our dataset after disassembling them, counting the number of tokens generated and summing over the entire dataset. Relative to the large size of our dataset in text form, approximately 2.8TB, we end up with comparatively few tokens. We believe this is primarily due to the very formulaic nature of IR where there are many long character sequences that will occur often enough to be tokenized into a single token. We note that this is a very naïve method of tokenizing a language as structured as LLVM-IR but believe this serves as an appropriate estimate for the number of tokens that one could expect to obtain from our dataset.

## 5 Downstream Usecases

ComPile is designed to accomodate a number of downstream use cases, which can broadly be categorized into compiler-related tasks, and the enhancement of datasets for foundation models for code. We will first present compiler-related downstream use cases, before proceeding to the use of ComPile in code models and their evaluations.

### 5.1 Compiler-related Tasks

There are a variety of compiler-related ML tasks that we believe ComPile to be useful for. Training machine learning models to replace heuristics to improve build times, code size, or performance has been done before, but typically using small, bespoke datasets (Trofin et al., 2021; Ashouri et al., 2022). Such heuristic replacements are typically structured as RL tasks where the agent makes a decision within the compiler and the reward is defined based

---

11. https://github.com/glample/fastBPE

on code size or runtime performance. These models perform significantly better, in some cases improving two-fold, with significantly more data .The limit to this trend is currently unexplored. In addition to the RL case, there are also several supervised tasks of great interest to the compiler community. Performance prediction has in particular been an active field of research (Li et al., 2022a; Abel and Reineke, 2022; Mendis et al., 2019). Previous datasets in this space (Chen et al., 2019) have often been relatively small and only available at a specific granularity. New performance datasets can easily be derived from ComPile as it is distributed in the form of unoptimized IR, which can be optimized, or compiled per the demands of the researcher. Large amounts of unoptimized IR also allows for effective testing of downstream tooling and additional data that augments ComPile, such as function inputs which allow for the code within ComPile to be executed (Ivanov et al., 2024). Making the code within ComPile executable allows for many further downstream tasks such as performance introspection, detailed execution analysis, and further research into utilizing execution data to improve ML techniques for code. All at a scale previously impossible, as the dataset collection presented too large of a barrier to overcome.

Finally, other supervised tasks, such as code size prediction, can also be useful in certain contexts. Code size prediction involves taking IR and predicting the resulting size of the `.text` section in the final object file, potentially after optimizations. While this use case is limited due to the relatively low cost of running the compiler to obtain the final cost, in some RL pipelines, being able to predict the final size without performing a full compilation is essential.

### 5.1.1 Design of Evaluation

To evaluate the quality of our dataset and compare against other compiler-focused datasets, we train a series of small LLMs to predict code size, both before (O0) and after optimization (O3). For an example pair of input IR, and target code-size please see Table 4. To create the code size fine-tuning dataset, we take only the C and C++ portions of ComPile, extract individual functions from the modules present, and then compile them to obtain the corresponding code size. We also collect the textual IR, tokenize it with a GPT tokenizer with a vocabulary of 32768, and discard any examples with a number of tokens that exceed the model's context length, in our case, 2048. We were able to collect approximately 2.7M functions from the C/C++ split of ComPile while rejecting 1.8M functions that did not fit into the context window. We obtained approximately 710k of the 1M functions within AnghaBench (Da Silva et al., 2021). In addition to fine tuning datasets, we prepare a pretraining dataset using the C/C++ portions of ComPile, chunking textual IR obtained from whole modules based on the models' context length. We compare against AnghaBench as it is the only compilable dataset known to the authors of similar scope. We process AnghaBench into LLVM Bitcode utilizing our dataset tooling and then further process it into a pretraining and a code size dataset using the exact same methodology and tooling we used for ComPile. To train models, we utilize the MPT architecture (Research, 2024) at sizes of 125M, 150M, 200M, and 250M parameters. Each model is pretrained for a specified number of batches depending upon the model size, coming out to approximately 20 tokens per parameter. The models are subsequently finetuned for the same number of batches on code size prediction. A subset of 1000 batches is used as a test set during training.

| Language | Unoptimized IR | Code Size | Optimized IR | Code Size |
|----------|----------------|-----------|--------------|-----------|
| C | ```define i32 @sum`
`(i32 %0, i32 %1) {`
`  %3 = alloca i32`
`  %4 = alloca i32`
`  store i32 %0, ptr %3`
`  store i32 %1, ptr %4`
`  %5 = load i32, ptr %3`
`  %6 = load i32, ptr %4`
`  %7 = add nsw i32 %5, %6`
`  ret i32 %7`
`}``` | 60 Bytes | ```define i32 @sum`
`(i32 %0, i32 %1) {`
`  %3 = add nsw i32 %1, %0`
`  ret i32 %3`
`}``` | 4 Bytes |

Table 4: Evaluation pairs in C for the task of code-size prediction on unoptimized IR (O0), and the code-size prediction on optimized IR (O3). The model sees the unoptimized, respectively optimized IR, and has to predict the code size in bytes.

To evaluate the models, we create an evaluation dataset using the LLVM test suite [12]. The LLVM test suite contains a variety of code snippets that are useful for evaluating compile time, compiler correctness, performance of generated code, and the size of generated code. We create the evaluation dataset using our dataset tooling and then process it into a code size dataset using the same tooling and methodology used to construct the training dataset. This is used to evaluate each model against the dataset. Two different code size prediction tasks are evaluated, particularly code size prediction with no optimizations, and code size prediction after the O3 optimization pipeline has been run. In both cases, the models are given the same input, the IR module before any optimizations have been applied. O3 code size prediction is a significantly more difficult task as not only does the model need to predict how each instruction contributes to the overall size given nuances like instruction selection and register allocation, the model also needs to predict how the optimization pipeline will mutate the function, which will often have a significant impact on the final size.

### 5.1.2 RESULTS

Results for O0 are shown in Table 5. We see significantly improved performance for the models trained on ComPile, with the mean absolute percentage error (MAPE) being 17.6% for the best model trained on ComPile versus 63.7% for the best model trained on AnghaBench. It is of note that no model performs particularly well on this evaluation dataset. While we have not performed experiments to confirm, we believe this is due to the nature of the LLVM test suite. It contains a variety of different code, a large portion of which will look significantly different than typical as it is designed to exercise and test specific compiler features. Results for O3 code size prediction, a more difficult task, are shown in Table 6. Again, we see significantly improved results for the models trained on ComPile, with the best performing ComPile-trained model achieving 52% MAPE while the best performing AnghaBench model only achieved 88.9% MAPE. However, we observe much more variable performance from the models trained on AnghaBench. We hypothesize that this is due to AnghaBench not containing many esoteric code patterns found within the LLVM test suite, leaving performance more to luck of the draw.

---

12. https://github.com/llvm/llvm-test-suite

| Transformer | ComPile | | AnghaBench | |
| | MAPE | Average Difference | MAPE | Average Difference |
| (m) | (%) | (bytes) | (%) | (bytes) |
| --- | --- | --- | --- | --- |
| 125 | 37.8 | 12.1 | 383.5 | 189.9 |
| 150 | 23.3 | 12.3 | 63.7 | 39.5 |
| 200 | 18.6 | 9.5 | 132 | 58 |
| 250 | 17.6 | 8.6 | 79.1 | 25.5 |

Table 5: Evaluation results for ComPile and AnghaBench for code size prediction at O0 on the LLVM test suite.

## 5.2 Improvement of Code-related Tasks by utilizing Compiler Infrastructure

The usages of ComPile we envision for code LLMs, and code LLM Agents, can be classified into two, in parts overlapping, categories: the enhancement of code translation with intermediate representations, and the development of new evaluations for LLMs, and LLM agents. Pioneered by TransCoder-IR (Szafraniec et al., 2023), IR has the potential to be useful for translation between programming languages, with IR being used as an intermediate translation step to, for example, translate from C++ to Rust. In addition, LLVM IR has been adopted by several code LLMs (Lozhkov et al., 2024; Zhu et al., 2024). Utilizing shared compilation infrastructure, this crop of LLMs is able to better extrapolate to low-resource programming languages built atop the same compilation infrastructure, such as Chapel. Datasets enhanced with ComPile hold the potential to enable a leap in performance for these models, by pretraining on ComPile with a fill-in-the-middle, or masking objective, see Table 7, and hence imbuing LLMs with a better understanding of LLVM IR. A further exciting avenue of research enabled by ComPile is the enhancement, and development of new LLMs, and LLM agent evaluations. ComPile enables the automatic generation of function inputs (Ivanov et al., 2024), opening up entirely new avenues to LLM, and LLM agent evaluations. For example, CRUXEval Gu et al. (2024), a recent evaluation suite, has to rely on 800 Python functions consisting of hand-designed input-output pairs for input prediction

| Transformer | ComPile | | AnghaBench | |
| | MAPE | Average Difference | MAPE | Average Difference |
| (m) | (%) | (bytes) | (%) | (bytes) |
| --- | --- | --- | --- | --- |
| 125 | 85.5 | 19.9 | 914 | 180.8 |
| 150 | 53.1 | 17.1 | 88.9 | 31.5 |
| 200 | 50.2 | 15.8 | 479.3 | 83.8 |
| 250 | 52 | 15.5 | 268.6 | 39.3 |

Table 6: Evaluation results for ComPile and AnghaBench for code size prediction at O3 on the LLVM test suite.

| Fill-in-the-Middle Objective | | Masking Objective | |
|---|---|---|---|
| **Input IR** | **Target IR** | **Input IR** | **Target IR** |

```
define void @swap          define void @swap
(ptr %0, ptr %1) {         (ptr %0, ptr %1) {
  %3 = alloca ptr            %3 = alloca ptr
  %4 = alloca ptr            %4 = alloca ptr
  %5 = alloca i32            %5 = alloca i32
                            store ptr %0, ptr %3
                            store ptr %1, ptr %4
                            %6 = load ptr, ptr %3       define void @swap          define void @swap
                            %7 = load i32, ptr %6       (ptr %0, [mask]) {         (ptr %0, ptr %1) {
                            store i32 %7, ptr %5         %3 = load i32, [mask]       %3 = load i32, ptr %0, align 4
                            %8 = load ptr, ptr %3        store i32 %3, ptr %1, [mask]  store i32 %3, ptr %1, align 4
                            %9 = load i32, ptr %8        store i32 %3, ptr %0, [mask]  store i32 %3, ptr %0, align 4
                            %10 = load ptr, ptr %4       ret [mask]                 ret void
  store i32 %9, ptr %10      store i32 %9, ptr %10      }                          }
  %11 = load i32, ptr %5     %11 = load i32, ptr %5
  %12 = load ptr, ptr %3     %12 = load ptr, ptr %3
  store i32 %11, ptr %12     store i32 %11, ptr %12
  ret void                  ret void
}                          }
```

Table 7: Pairs for fill-in-the-middle, and masking tasks, often used for pretraining, on ComPile with a fill-in-the-middle objective on an unoptimized IR example, and a masking objective on an optimized IR example.

and output prediction tasks. ComPile enables the complete removal of this benchmark size constraint. At the same time, compiler IRs are slowly seeping into ever more LLM evaluations, with StarCoder 2 Lozhkov et al. (2024), and DeepSeek-Coder 2 (Zhu et al., 2024) both making use of limited LLVM IR datasets. Evaluations seeking to capture more facets of programming have begun to place a growing emphasis on code execution for LLM, and LLM agent benchmarks (Wang et al., 2022; Olausson et al., 2023; Gu et al., 2024; Jain et al., 2024). Compilation is an integral part of program execution, and ComPile offers a large corpus of compiled programs, whose latter stages of execution can be captured from the stage of the unoptimized LLVM IR distributed by ComPile. A nascent avenue of research is the reasoning about code execution with chain-of-thought (Wei et al., 2022) traces (Ni et al., 2024), here ComPile provides an essential building block to reason about execution traces at a much larger scale as ComPile's programs are guaranteed to compile and with appropriate tooling, can easily be made executable.

## 6 Related Work

There exist a number of related datasets of code for the training of machine learning models in literature. Conceptually, we break these related datasets down into three main categories, as shown in Table 8. Case 1 consists of datasets translating between two different codebases, case 2 considers reference work which translates between two different languages by going through the IR as an intermediate translation step, case 3 consists of a dataset of different languages without the structure to translate from one language, to the other explicitly, and case 4 consists of a number of source languages, compiled to the IR, which, to our knowledge, only contains our dataset, ComPile, and the dataset used in the work of Cummins et al. (2023). Most pretraining datasets for large language models (Li et al., 2022b; Kocetkov

| Name of Dataset | Size (TB) | Programming Languages | Case |
|---|---|---|---|
| The Stack | 2.9 | 358 Languages | Case 3 |
| The Stack v2 | 32.1 | 358 Languages | Case 3 |
| ComPile (closed) | 2.4 | Rust, Swift, Julia, C/C++ | Case 4 |
| ComPile (public) | 1.9 | Rust, Swift, Julia, C/C++ | Case 4 |
| Code Llama | 0.86 | ≤ 358 Languages | Case 3 |
| TransCoder | 0.74 | C++, Java, Python | Case 1 [3] |
| AlphaCode | 0.72 [1] | 12 Languages | Case 3 |
| LLM for Compiler Opt. | 0.001 | C/C++ | Case 4 |
| TransCoder-IR | [2] | C++, Go, Java, Rust | Case 2 |
| HPCorpus | 0.07 | Fortran, C, C++ | Case 3 |

Table 8: Related datasets to our newly introduced dataset are the Stack (Kocetkov et al., 2022), the Stack v2 (Lozhkov et al., 2024), the datasets used for the training of Code Llama (Rozière et al., 2023), TransCoder (Lachaux et al., 2020), AlphaCode (Li et al., 2022b), LLMs for Compiler Optimization (Cummins et al., 2023), and the HPCorpus (Kadosh et al., 2023b). Code Llama, as well as AlphaCode, use filtered subsets of GitHub Activity Data, where the filtering criteria of Code Llama are not known. We break all related datasets down into 4 distinct cases: **Case 1:** The translation between two programming languages, **Case 2:** The translation between two programming languages through the intermediate representation as an intermediate step, **Case 3:** A mix of different codebases from different programming languages, and **Case 4:** A mix of different codebases from different programming languages compiled to the intermediate representation.

et al., 2022; Lozhkov et al., 2024; Markovtsev and Long, 2018) fall into the third category, scraping source code from hosting services like GitHub, and GitLab with their expansive index of individual repositories in all programming languages, and hence tend to produce very extensive datasets which are only filtered for licensing issues, and then deduplicated, but do not take the quality of the included code into account.

Relating to our dataset, datasets from the third class also do not guarantee that they, in themselves, are compilable, and often contain auxiliary files such as documentation in Markdown. Another favored source of code within this category is programming competitions, and while such code is inherently compilable, it bears little resemblance to code used in production. In the case of modern large language models, the quality of the code is mitigated by only using the data for fine-tuning (Brown et al., 2020), or further instruction-tuning with reinforcement learning (Ouyang et al., 2022) to achieve the desired downstream behaviour.

Case 1 contains a number of recent datasets for models which transcode between two programming languages, examples of which include Transcoder (Roziere et al., 2020), and

---

1. This figure only includes the pretraining dataset for AlphaCode rather than the smaller competition sourced fine-tuning dataset.
2. Size of training dataset not reported, and not reproducible. Dataset consists of 9.5B tokens, tokenized with fastBPE (Sennrich et al., 2016).

recent efforts like the one of IBM to translate COBOL to Java watsonx (2023). Depending upon the specific methodology used for training, datasets for this case can look similar to datasets for case 2 when techniques like back-translation are employed for model training, but we make the distinction here primarily on dataset usage. The extension of case 1 to translate between two different programming languages by utilizing the intermediate representation as an intermediate translation step transforms such a dataset into Case 2. The only example of this known to the authors is Transcoder-IR (Rozière et al., 2023). Complementary to these large pretraining-scale datasets, there exist a number of smaller, more focussed datasets aimed at the fine-tuning of already pretrained large language models (Zhu et al., 2022; Li et al., 2022b; Puri et al., 2021). These datasets are primarily collected through data extraction from coding competitions (Li et al., 2022b; Puri et al., 2021), or the scraping of curated websites (Zhu et al., 2022). This guarantees a higher level of quality in regards to buildability and structure for the included code, hence making them more optimal for fine-tuning. However, the data collection methodology implicitly introduces a lack of variety in the datasets. Coding competititon datasets might include a couple thousand coding exercises which contain a great many solutions to the same exercises, but yet they are only solving the very same set of coding problems. Optimizing for time-to-solution or other narrow properties, such code also exhibits decidedly different characteristics to code used in production, hence making these datasets markedly different to ComPile. Specifically for the task of machine-learned compiler heuristics, and machine-learned compiler componentry there exist a number of statistics-focussed (Kadosh et al., 2023a), compiler heuristics-focussed (Armengol-Estapé et al., 2022), and autotuning-focussed datasets (Da Silva et al., 2021). Often beginning with the web-scraping of large amounts of code, these approaches modify the resulting code in a number of ways. Examples include the modification of arbitrary source files to make them compilable (Da Silva et al., 2021), executable (Armengol-Estapé et al., 2022), or abetting the statistical analysis of aspects of the code (Kadosh et al., 2023a). ComPile, while being able to fulfill similar dataset demands, offers a number of key advantages. The code in our dataset, by means of our dataset construction methodology, consists only of compilable code, using the same compilation toolchain as used for production deployments, of which the IR is collected before optimization, allowing for IR at any stage of the compilation pipeline to be easily generated. This allows ComPile to go significantly beyond the capabilities of previous compiler-targeted datasets.

## 7 Limitations and Future Work

The presented dataset introduces a large corpus of compiled high-quality code. While this work has very good coverage of languages such as Rust, Swift, and Julia, we had to make a number of implicit trade-offs in the construction of our dataset. Compared to a number of other larger datasets such as the Stack 1 & 2 (Kocetkov et al., 2022; Lozhkov et al., 2024), we decided to not pursue a number of avenues to obtain the same order of magnitude of tokens, opening up a number of future avenues of work. Following our approach to only include high-quality code in our dataset, we believe the dataset could be significantly expanded by taking advantage of additional package ecosystems such as those of Linux distributions. These ecosystems contain recipes with a consistent format across all packages that describe the individual building steps and dependencies. Some individual package ecosystems contain

close to 1M recipes (AUR, 2023), and could hence prove fruitful to the expansion of our IR database. Adopting the widely used approach of GitHub repository scraping, we could also envision filtering the list of repositories compilable with LLVM-IR generation, adding to the corpus only those repositories of proven quality. Filtering parameters such as the number of contributors, number of commits, and other activity metrics could remove outlier repositories that do not contain high-quality code (e.g., homework assignments and abandoned code). However, we view this task as challenging due to these repositories not abiding by a consistent format, having much higher variance in their build systems, and hence being much harder to compile with a single builder. This problem extends to the dependencies of said GitHub repositories. Not all the projects have a consistent dependency specification, hence requiring either manual dependency resolution or resolution in a highly complex automated fashion.

Furthermore, we did not include a number of other language-specific ecosystems due to the inherent difficulties and fragmentation of their build systems. Haskell, for example, builds on the LLVM compilation infrastructure and has a centralized package repository with specified dependencies. Nevertheless, including it in this dataset proved infeasible due to the complexity of Haskell's dependency management which requires very specific Haskell versions and highly specific dependencies with pre-specified versions. These convoluted dependencies are almost impossible to be handled by the central builder schemes used in this project. Other languages, like the HPC-language FORTRAN, did not see wider inclusion in the dataset due to the varied compilation behavior of its multiple LLVM frontends. Added complexity came from highly specific compilation flags for each FORTRAN project which vary from compiler frontend to compiler frontend and from build system to build system, hence making it impossible to compile with one central compilation approach. While we could explore the integration of code from other datasets such as the Stack 2 (Lozhkov et al., 2024), coding competition datasets (Li et al., 2022b), and HPC-focussed datasets of code (Kadosh et al., 2023a), these would still be subject to the same restrictions as in the preceding paragraph. They would have to be filtered for code quality, their build systems inspected for the amenability to a centralized builder, and for compiled languages such as C/C++ and FORTRAN their compilation would have to utilize a LLVM-based compiler.

In future work, we seek to expand upon our approach to more closely align large language models for code with compilation infrastructure. We will explore code-centric tokenization as an opportunity to closely incorporate knowledge about the programming language and IR structure. We believe that by departing from textual tokenization of the IR, as is beginning to be explored in the literature (Guo and Moses, 2022; Szafraniec et al., 2023), we can provide improved performance over the current state-of-the-art. Our primary motivations for this belief include the fact that the distance between certain elements of IR, like attribute groups, and relevant context, like their associated definitions, can easily exceed the context length of most LLMs, in addition to the possibility for more compact tokenizations allowing more context to be given to models. In using domain-specific tokenizations, we hope to preserve more of the semantic structure of the IR throughout the tokenization, and hence improve downstream model performance. This approach will hopefully be able to yield small, performant models, which also retain the performance of larger models trained on textual tokenization. In follow-up work, we seek to further explore the statistical properties of the dataset such as the distribution of code within the dataset, the impact on the distribution

of code by different collection methods, and the performance impact of trained models these different distributions end up having. Further influences to be quantified are the influence of the dataset construction techniques, the influence of the sources of the dataset on the distribution of the dataset, and most importantly the impact of distribution shifts in code datasets on downstream model performance.

With a fast moving project like LLVM, there are often significant changes between versions. Even within the rendition of the dataset presented here, there is bitcode from multiple versions of LLVM depending upon the specific frontend used to compile a piece of code. Many of these updates to LLVM involve significant changes to how IR should be produced by a compiler frontend, such as the change from typed to opaque pointers. Having a static dataset means that the dataset becomes less relevant over time as LLVM evolves further. We publish our tooling to produce the dataset which allows for the creation of a dataset similar to this one produced with arbitrary frontend versions and leave it to future work to quantify the impact that distribution shifts over time might have on the utility of a dataset such as this one. Beyond the training of large language models, we also envision extensive future use of the dataset for the training of machine learned compiler components. First, machine learned compiler heuristics have shown great promise (Trofin et al., 2021), but are held back by limitations of current datasets. ComPile enables the metrics to be better trained, which has shown ample performance improvement in practice with performance metrics sometimes doubling from having access to large amounts of previously hard to gather IR. Going beyond the improvement of existing machine learned compiler heuristics, the presented dataset could furthermore make the training of heuristics such as e.g. the inlining-for-size in generic cases much easier. While learned compiler heuristics only touch individual stages of the compilation pipeline, ComPile enables much more far-reaching work on performance evaluations of LLMs as models of entire compilation pipelines, which has only been explored on small datasets previously (Guo and Moses, 2022; Mannarswamy and Das, 2022). ComPile is much broader in scope than the previously tested datasets, and our data collection approach allows for the collection of IR at *any* point of the compilation pipeline through simple postprocessing pipelines, hence enabling entirely new avenues of LLM compilation pipelines research.

The developed dataset collection, and compilation tooling is also to be further explored in future work. Its extensible pipeline could potentially be used to automatically execute a plethora of unit tests and benchmarks throughout the build system, and hence much better verify the individual stages of compilation and IR transformations. Extended with function instrumentation, and replay tooling (Castro et al., 2015), future derivatives of ComPile could also include function inputs, and expected outputs along with extracted functions to allow for fine-grained performance introspection on a grand scale, which is currently impossible. Potential future results of such fine-grained introspection throughout the compilation pipeline, and across programming languages include better performance prediction without needing to compile a single line of a program, better evaluation of the performance impact on individual compiler optimizations, and performance improvement through better compiler-generated code.

## 8 Conclusion

In this paper we present ComPile, a novel dataset of LLVM-IR collected from a number of package ecosystems consisting of large production-grade codebases. It is significantly larger than previous finetuning-focussed, and compiler-focussed code datasets, albeit smaller than large language model-focussed code pretraining datasets. Statistical analysis of the collected dataset is performed, and differences in the IR properties of the collected IRs between languages are being shown. ComPile's increased size in combination with its quality-focused construction methodology not only enables the systematic evaluation of previous work, but opens up entirely new avenues of research for IR-centric machine learning, and most specifically machine-learned compiler componentry for which the scale of this dataset paves the way to an entirely new generation of machine learning models for compilers.

## Broader Impact Statement

The ComPile dataset will have the largest impact in machine learning for compilers, where it constitutes the first large language model scale dataset of compiler representation. It will hopefully enable wider progress in the application of machine learning to compilers, and the design thereof.

The potential uses of the dataset may benefit a large user base due the ubiquitous use of LLVM across compilers in industry such as the ones from Apple, Intel, IBM, and AMD, as well as LLVM's just-in-time (JIT) compiler, which used by a number of programming languages such as Julia, and the widely-used Python. Each optimization, heuristic, or learned pass ordering will in some capacity apply to all of these, and hence be for the broad benefit of all. While the, repeated, building of ComPile snapshots leads to a short-term increase in greenhouse gas emissions, the exact amount is hard to quantify due to varying datacenter efficiency. This is expected to be offset by the long-term benefits brought about by better compiler heuristics, and machine learning-improved compiler infrastructure trained on ComPile, whose impact is compounded by the widespread use of the LLVM compiler infrastructure.

There are important considerations made in the construction of ComPile to respect the licenses of the software packages ComPile is built from. The dataset is filtered for permissive licenses, as outlined in subsection 3.5, and licenses are distributed alongside with the dataset. In addition, the public release of the dataset went through the rigorous internal release review of Lawrence Livermore National Laboratory (LLNL).

We would encourage further work into the biases inherent to the dataset, and its internal distribution of intermediate representation sources. Its construction is a conjunction of our best effort to represent the wider usage of LLVM across programming languages, and the ability to extract intermediate representation from centralized package indices. As such it is not representative of the wider usage of LLVM as outlined in section 7. To ensure the long-term benefit of ComPile, it's representative evaluation of the usage across languages

utilizing the LLVM compiler infrastructure is going to be of the utmost importance. The exact impact of this is an open research question.

## Acknowledgments and Disclosure of Funding

We would like to thank Valentin Churavy for his assistance in understanding the Julia-compiler, and for volunteering his in-depth knowledge on pathways to the extraction of IR from Julia packages. We would also like to thank Todd Gamblin, Alec Scott, Harmen Stoppels, and Massimiliano Culpo for their assistance with Spack, and their prompt reviewing of our changes to Spack. We would furthermore like to express our gratitude to Nikita Popov, Arthur Eubanks, and all other LLVM contributors who helped with the reviews of the required patches to upstream LLVM.

The views and opinions of the authors do not necessarily reflect those of the U.S. government or Lawrence Livermore National Security, LLC neither of whom nor any of their employees make any endorsements, express or implied warranties or representations or assume any legal liability or responsibility for the accuracy, completeness, or usefulness of the information contained herein. This work was in parts prepared by Lawrence Livermore National Laboratory under Contract DE-AC52-07NA27344 (LLNL-JRNL-854809).

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

## Appendix A. Dataset Statistics of the Closed Dataset

| | | | BPE Tokens | | | |
|---|---|---|---|---|---|---|
| **Language** | | **Llama 2 Tokens** *(billions)* | **10k** *(billions)* | **50k** *(billions)* | **50k** *(billions)* | **100k** *(billions)* |
| | C | 16 | 3 | 2 | 2 | 1 |
| | C++ | 116 | 30 | 17 | 14 | 12 |
| julia | Julia | 615 | 48 | 27 | 20 | 14 |
| | Rust | 1079 | 198 | 132 | 116 | 102 |
| | Swift | 21 | 4 | 2 | 1 | 1 |
| **Total** | | **1848** | **282** | **179** | **154** | **130** |

**Closed Dataset**

Table 9: Token count of the encoded ComPile under varying vocabulary sizes, and considering the tokenization of the data with Byte-Pair encodings *(BPE)*, and tokenization with the Llama 2 tokenizer.

