# OpenReview forum: "ComPile: A Large IR Dataset from Production Sources"
_DMLR — Accepted by DMLR_

### Review · Reviewer_KsPN · 2024-05-30

**Recommendation:** 3
**Confidence:** 1

**Summary Of Contributions:**

Recent LLMs that are trained on code have unlocked capabilities such as code generation and evaluation. Currently, the code datasets used to train these models tend to be scraped from open source repositories on Github across numerous programming languages. One important property of code that such a data curation process ignores is that there are common structures in the code across all language. In particular, when code is compiled, there is an intermediate representation (IR) that is language-agnostic. This paper thus explores how to produce a dataset of IR code samples spanning numerous programming languages, with the goal of training a ML model on high-quality IR data for compilers. They provide language-specific strategies on how to obtain IRs as well as how to deduplicate samples. Preliminary analysis on the distribution of IR instructions, dataset size, and tokenization is discussed.

**Strengths:**

* Utilizing intermediate representations is an understudied technique to produce more structured, high quality code datasets for machine learning tasks involving compilers.
* Paper provides thorough detail in describing the per-language techniques of how to construct the IR data as well as filtering strategies.

**Audience:**

Yes

**Claims And Evidence:**

Claims are well-supported.

**Datasets And Benchmarks:**

Detail on data collection and organization was sufficient.

**Extended Submissions:**

N/A

**Limitations:**

* Clarity: It was not clear what downstream tasks a model trained on this IR dataset should accomplish. While there was some mention of use cases such as heuristics and compiler pass ordering, they were not formally defined. Moreover, the paper made many references to code datasets for language models (such as TransCoder and The Stack), which was confusing and made me initially expect that the dataset was created to train better code language models, not ML models for compilers.

* Quality: while the paper provides statistics on the dataset, there are no preliminary results on how a model trained on this dataset (even fine-tuned) performs on desired tasks. It is important to benchmark this dataset---even with an imperfect model---to establish how it can be used in future ML research.

Edit after author response: Thank you for the examples of downstream tasks and the results on code size prediction. I have updated my score.

**Requested Changes:**

* Provide formal definitions of downstream tasks; for instance, "Task name: .... Input: [partial IR sequence] Output: [instruction]"

**Strengths And Weaknesses:**

See Strengths and Limitations.

---

### Review · Reviewer_hFgA · 2024-06-03

**Recommendation:** 4
**Confidence:** 1

**Summary Of Contributions:**

The authors present a dataset called ComPile, which consists of a 2.8TB collection of textual LLVM intermediate representations (LLVM-IR). The dataset contains IRs from high-quality ecosystem libraries in Rust, Julia, Swift, and C/C++. The dataset is designed to be large-scale to be suitable for pre-training of large language models specialized on LLVM-IR. The authors provide a statistical analysis of the dataset and open-source the workflow to reproduce and extend the dataset.

**Strengths:**

Novelty and Uniqueness: The ComPile dataset is the largest of its kind, offering a dataset of compiler representation at language model scale.

Quality-Focused Construction: The dataset's construction methodology prioritizes compiling high-quality code using the same toolchain as production deployments. This emphasis on quality ensures that the dataset contains reliable and valuable data for research and development.

**Audience:**

Yes

**Broader Impact Concerns:**

No concerns.

**Claims And Evidence:**

Yes, but see limitations and requested changes.

**Datasets And Benchmarks:**

Yes. The dataset was published as well as the code to reproduce and extend it.

**Extended Submissions:**

No previously published peer-reviewed work found.

**Limitations:**

- have been addressed in revision

**Requested Changes:**

Minor remarks:
- Barplots in Figure 3 are not well-interpretable. x-Scale should not show discrete numbers, but a range (?)

**Strengths And Weaknesses:**

S: The dataset is the largest of its kind and the authors made a great effort in using high-quality and permissively licensed input, as well as creating a reproducible and extensible code->LLVM-IR stack that can be build upon in future work.
S: Interesting downstream use cases, supported by promising experimental results on two downstream use cases.

---

### Review · Reviewer_GQC8 · 2024-06-04

**Recommendation:** 4
**Confidence:** 2

**Summary Of Contributions:**

This paper proposes a large IR dataset from production sources, called ComPile, which aims at enabling the training of larger language models for compilers useful for a broader array of downstream tasks after fine-tuning, and in the long-term enabling use-cases to obtain a reliable runtime estimate without needing to run a single line of code. It sounds pretty ambitious to me. The main contributions include the dataset collection and preliminary statistical analyses.

**Strengths:**

See above.

**Audience:**

Yes

**Broader Impact Concerns:**

Broader impact is carefully discussed in the paper and I have no concerns on possible negative broader impact.

**Claims And Evidence:**

Claims are well support by evidence in this paper.

**Datasets And Benchmarks:**

Limited information on maintenance and responsible use. See Weaknesses.

**Extended Submissions:**

N/A

**Limitations:**

Limitations are carefully discussed in the paper and I have no concerns on those.

**Requested Changes:**

See Weaknesses.

**Strengths And Weaknesses:**

Strengths:
1. The paper definitely fits the scope of DMLR by introducing a new dataset which could be useful for LLM training for compilers.
2. Overall, the paper is well organized and written. Detailed information is provided on dataset construction and availability.
3. License filtering is carefully used to ensure proper use of package repositories.

Weaknesses:
1. More information on dataset maintenance is needed. Can future work be written as a development plan? Do authors encourage more online collaboration of dataset maintenance?
2. More information on responsible use of dataset is needed. In Section 4, there is only a subsection showing some demonstrations of utility of ComPile for training LLM through quantification of size and approximate token counts. However, one may want to know how they can use the dataset in general. In detail, they want to know how to apply existing methods, and what results can be generated to show why ComPile is better than related datasets.